# Titanium Carbide and Vibration Effect on the Structure and Mechanical Properties of Medium-Carbon Alloy Steel



**Tatyana Kovalyova [1], Yevgeniy Skvortsov [1,*], Svetlana Kvon [1], Michot Gerard [2], Aristotle Issagulov [1], Vitaliy Kulikov [1] and Anna Skvortsova [3]**

1   Nanotechnologies and Metallurgy Department, Karaganda Technical University, Karaganda 100012, Kazakhstan; sagilit@mail.ru (T.K.); svetlana.1311@mail.ru (S.K.); aristotel@kstu.kz (A.I.); mlpikm@mail.ru (V.K.)
2   Jean Lamure Institute, University of Lorraine, 54011 Nancy, France; gerard.michot@univ-lorraine.fr
3   Obstetrics, Gynecology and Perinatology Department, Non-Profit Joint-Stock Company Karaganda Medical University, Karaganda 100000, Kazakhstan; doktor-anna@mail.ru
*   Correspondence: yev.skvortsov@gmail.com

**Abstract:** This study aimed to improve the hardness and wear behavior of medium-carbon alloy steel through the addition of titanium carbide ultradispersed powder and low-frequency vibration treatment during solidification. It was shown that the complex effect of low-frequency vibration with the additional introduction of a small amount of titanium carbide ultradispersed powder with the size of 0.5–0.7 μm during the casting process had a positive effect on structural changes and led to improved mechanical properties, and so increasing the value of microhardness by 37.2% was notable. In the process of shock dynamic impact, imprints with crater depths of 13.69 μm (500 N) and 14.73 (700 N) were obtained, which, respectively, are 23.34 and 42.34% less than that on the original cast sample. In the process of tribological testing, decreasing the depth of the wear track (50.25%) was revealed with decreasing the value of the friction coefficient by 14.63%.

**Keywords:** wear resistance; microhardness; abrasion test; tribological test; microstructure

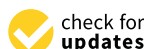



## 1. Introduction

An increase in the volume of ore production at the deposits with a simultaneous constant decrease in the presence of the necessary metals in the initial mineral resources and the requirements for reducing the fineness of grinding in order to fully and comprehensively use raw materials determine the need to improve the quality and increase the productivity of equipment reliability. Durability is directly related to such an indicator of the work of a huge number of machines and mechanisms as the wear resistance of parts. The increase in the resource of effective operation of equipment during impact-abrasive wear is achieved by obtaining the optimal parameters of the structure and the operational properties of cast products. However, castings or blanks obtained by casting have defects such as shrinkage cavity, shrinkage porosity, and rough structure, which lead to the deterioration of operational characteristics [1]. In view of these problems, casting researchers are conducting a large amount of scientific research aimed at clarifying the structure of the casting or work piece and improving its mechanical properties.

It is known that the germinal submicron particles (including nanoscale ones) of refractory carbides, oxides, sulfides, nitrides, and borides are introduced from the outside into metals and alloys of various compositions to actively influence the processes of crystallization and structural transformation [2,3]. Ti-based coatings reinforced with TiC were obtained on the $Ti_6Al_4V$ surface using a mixture of $Ti_6Al_4V/NiCr-Cr_3C_2/Y_2O_3$ powders in a research paper [4]. The results showed that the cracks were completely eliminated, and the number of pores was drastically reduced with the addition of 2 wt.% $Y_2O_3$. In addition, the hardening TiC and matrix layered solution β-Ti were the main deposited phases inside the coating. A study was conducted to evaluate the effect of adding $Al_2O_3$ particles in an

amount of 0.5 wt.% with an average particle size of 500 nm (0.5 microns) on the mechanical properties and wear resistance of the AISI 316 austenitic stainless steel matrix. It was found that the introduction of $Al_2O_3$ particles increased the hardness of austenitic stainless steel from 161 to 185 HV1, but reduced the impact strength from 160 to 143 J [5]. During other research, modification of a continuous carbon steel ingot was carried out using TiN, $Y_2O_3$ powder modifiers in the amount of 0.03–0.05 wt.%. The analysis of the structure showed a significant change in the morphology and dispersion of crystalline grains. Instead of a coarse dendritic structure, a globular dispersed crystal structure was formed. The physical and mechanical characteristics of the metal changed significantly. The central porosity decreased by 25–36.7%, the total chemical heterogeneity by 39.8–75%, liquation and total fracturing by 34–100%, the zone of equiaxed crystals increased by 26.5–35%, and the mechanical characteristics of the metal (tensile strength and yield strength, elongation and compression) increased by 5.5–19%. Castings from GX120Mn13 steel were obtained, which were cast into sand molds at atmospheric pressure using powder modifiers in the form of a TiN+Cr composition. Changes in the morphology and grinding of crystals in the sample after modification were observed. The degree of grain grinding was 18.7%. The modified samples had a reduced mass loss when testing the samples for wear resistance. Studies of P265GH steel have been conducted. The steel was modified with TiCN particles in the amount of 0.1 wt.%. The results showed that the modification led to a decrease in the average diameter of the ferrite grain from 112 μm to 50 μm and to a more uniform grain size distribution. In another study, a TiCN+$Y_2O_3$+Ni+Fe nanomodifier was introduced into the melt as part of the composition: TiCN:$Y_2O_3$:Ni:Fe = 2:1:1:6. Here, Ni and Fe were cladding metals. In the unmodified sample, the graphite phase had the basic classical shape of elongated plates. In the modified version, it formed a grid in the form of club-shaped and rounded small inclusions. It was established that the conditional average diameter of these inclusions for unmodified samples was 181 μm, and for modified samples 28 μm. Comparative studies of the mechanical characteristics of cast iron ingots show an increase in yield strength by 9%, tensile strength by 15%, elongation by 36%, and wear resistance for castings with ultradispersed powders by 16% [6]. The experience of using ultrafine particles is known from the practice of hardening heat-resistant alloys [7] and aluminum alloys [8,9]. There are also positive examples of the use of various ligatures of rare earth metals and alloys based on them being used as modifiers [10]. The analysis of the information shows that the introduction of ultrafine powder modifiers significantly improves the mechanical properties of alloys, including wear resistance.

However, some problems related to the peculiarities of introducing particles into a liquid iron–carbon melt, the physicochemical characteristics of the injected particles, as well as their stability in liquid and hardening metal under conditions of no equilibrium crystallization, remain unresolved at the moment [11,12]. At the same time, the use of ultradispersion in metallurgy is a very difficult issue due to the instability of ultrafine powder substances, since the tendency of the number of particles forming agglomerates increases significantly [13]. In order to destroy particle agglomerates, improve particle wetting, and ensure uniform particle distribution, it is necessary to use some external factors in the melt volume. Acoustic fields (from ultrasonic to low-frequency) can be used as such external factors.

In accordance with modern ideas about the solidification of cast blanks, significant reserves for improving the quality of metal products can be found at the stage of transition from the liquid phase to the solid phase. This is due to the fact that, at this stage, the quality of future products and the level of physical and mechanical properties and operational characteristics are being formed. From this point of view, it seems very promising to use methods of forced mixing of the liquid phase during solidification, providing control of the mass transfer and heat exchange processes [14]. Due to the simplicity of technical implementation, low energy consumption, and high productivity (the ability to process a large volume of melt), the vibration processing method is of particular interest for controlling the formation of cast products during solidification, which provides directional and regulated

mixing of liquid flows and solid phase particles contained within them [15]. Some research associates the positive effect of vibration treatment on the structure of cast blanks with the cavitation mechanism [16]. Kishor Pawar and others have described a significant improvement in the microstructure and mechanical properties of aluminum using ultrasonic treatment. Castings with ultrasonic vibration had a microhardness value 12% higher than castings obtained without ultrasonic treatment [17]. Identical microstructural behavior was also described in [18,19]. It has been demonstrated that high-intensity oscillations eliminate large aluminum alloy grains and form spherical equiaxed grains with a size of about 25 μm [20]. Additionally, Limmanevich and his colleagues illustrated changes in the structure of cast aluminum alloys as a result of low-frequency vibration effects. It was found that the average grain size of the primary phase became relatively smaller and spherical as the frequency of vibration increased [21]. The optimal frequency (60 Hz) and amplitude (0.5 mm) of the vibration treatment of the hardening aluminum alloy A356 were experimentally found, which led to a significant reduction in the grain size in the ingot, as well as an increase in the strength (density) of the metal in the finished form. The yield strength increased by about 80% compared to the ingot cast without vibration [22,23]. Changes in the parameters of plasticity, toughness, and hardness of the Mg-Al-Zn alloy were demonstrated in the scientific work [24] as a result of a change in the vibration frequency. It was also found that aluminum–silicon alloys LM25 (Al–Si 7.15%) and LM6 (Al–Si 12.30%) obtained via the vibration of a metal mold had lower porosity compared to casting without vibration [25]. In turn, it was shown in [26] that the introduction of diamond nanoparticles with simultaneous low-frequency vibration treatment with a frequency in the range of 10–100 Hz of the solidifying melt significantly reduces the average grain size (up to 0.5 mm) and leads to a significant improvement in the mechanical properties of the A356 alloy. Kai Qiu and others have illustrated microstructural changes under the influence of vibration in the process of producing gray cast iron by lost foam casting. It was found that, due to the effect of vibration during the solidification of the casting, the flow of the metal melt accelerates, the temperature gradient of the metal melt decreases, and fluctuations in the concentration of carbon atoms occur, which contributes to the homogenization of the composition of the metal melt so that the germ-like flake graphite cannot grow, and as a result, a flake of graphite of shorter length and smaller thickness is formed. Vibration treatment led to an increase in hardness, tensile strength, and elongation of gray cast iron after destruction [27]. It was found that, at a vibration frequency of 35 Hz, the microstructure of gray cast iron is denser than that of a sample without vibration, flake graphite of type A becomes thinner and shorter, and the effect of refining primary austenite is very obvious. Thus, it was possible to increase the complex of mechanical properties [28]. To assess the microstructural features, ductile iron EN-GJS-450-10 was manufactured with vibration with amplitudes of 0.9 mm and 1.8 mm at a fixed vibration frequency of 50 Hz. Cast samples and samples treated with vibration were compared. The percentage of ferrite and graphite areas significantly improved from 24% and 16.5% for the cast sample to 57% and 22.3% for castings after vibration with an amplitude of 1.8 mm, while the percentage of perlite and pores significantly decreased from 56.1% and 5% to 20% and 1%, respectively [29].

A review of scientific papers shows that vibration treatment leads to changes in the structure and mechanical properties. This is expressed not only in the creation of a high-quality structure in terms of metallurgical characteristics (absence of segregation, porosity, the quantity of nonmetallic inclusions), but also in terms of the quality of the microstructure: grain size and orientation, improved introduction and distribution of particles, as well as the degassing of the melt. Research on the effects of vibration on crystallizing melts is mainly focused on aluminum, siliceous, and magnesium cast alloys [16–26]. Although the positive effect of vibration treatment on cast products made of iron–carbon alloys has been shown in the technology of production of castings according to smelted models [27,28], no scientific papers have been found to study the effect of vibration on the structure and properties of cast steel alloys.

The purpose of this work is to study the effect of the introduction of ultrafine titanium carbide powder and low-frequency vibration treatment during primary crystallization on the structure and properties of 35HGSL steel, the chemical composition of which is shown in Table 1. The choice of 35HGSL steel as an object of research is associated with the widespread use of this alloy as a material for the manufacture of cast parts of metallurgical equipment operating under wear: conveyor sprockets, couplings, gears, shafts, and other critical parts that require increased wear resistance.

**Table 1.** Treatment modes for experimental samples.

| Sample Number | Characteristic of the Treatment Mode |
|---|---|
| 1 | cast sample of an experimental alloy, without vibration, without introduction of TiC |
| 2 | vibration: frequency 15 Hz, amplitude 0.5 mm. |
| 3 | TiC in the amount of 0.1% wt., fineness 0.5–0.7 microns |
| 4 | cast sample of an experimental alloy, without vibration, without introduction of TiC |
| 5 | vibration: frequency 30 Hz, amplitude 0.5 mm. |
| 6 | TiC in the amount of 0.2% wt. with fineness 0.5–0.7 microns |
| 7 | cast sample of an experimental alloy, without vibration, without introduction of TiC |
| 8 | vibration: frequency 45 Hz, amplitude 0.5 mm. |
| 9 | TiC in the amount of 0.3% wt. of the mass with fineness 0.5–0.7 microns |
| 10 | cast sample of an experimental alloy, without vibration, without introduction of TiC |
| 11 | vibration: frequency 60 Hz, amplitude 0.5 mm. |
| 12 | TiC in the amount of 0.4% wt. with a dispersion of 0.5–0.7 microns |
| 13 | cast sample of an experimental alloy, without vibration, without introduction of TiC |
| 14 | vibration: frequency 45 Hz, amplitude 0.5 mm and TiC 0.4 wt.% |

The choice of titanium carbide as an ultrafine modifier is related to the following circumstance. In the works under consideration, rare earth metal oxides are mainly used as ultrafine modifiers. They are quite expensive, which limits their use on an industrial scale. Consequently, characteristics of titanium carbide such as its chemical inertness, high hardness, and relatively low cost served as the basis for choosing it as an ultrafine modifier.

It should be noted that in the Republic of Kazakhstan, the full-mold process has become widespread as a casting method for medium-carbon alloy steels. The reserves for improving the operational properties of these steels have actually been exhausted; however, the results of the research of the above works suggest that the introduction of ultrafine modifiers and the use of vibration treatment will lead to structural changes and improve their operational characteristics.

## 2. Materials and Methods
### 2.1. Materials

To prepare the alloys, the following charges were used: steel scrap (Mechel, Moscow, Russia); slag mixture: lime CaO (65 wt.%), magnesite $MgCO_3$ (10 wt.%), fluorspar $CaF_2$ (25 wt.%); deoxidants: ferromanganese FMn75 SS 1415-93, aluminum AB92 SS 295-79; and alloying materials: ferrochomium FCh025B SS 4757-91, ferrosilicium FS45 SS 4755-91.

Titanium carbide grade CT 125/100, according to TU 3989-002-12606601-2006 (Thermosintez, Chelyabinsk, Russia), was used as an ultradispersed powder modifier.

### 2.2. Methods
2.2.1. Grinding of Titanium Carbide

Prior to use, the TiC fineness of the claimed 125/100 μm was reduced to 0.5–0.7 μm in a high-speed Emax-type nanoball mill (Retsch GmbH, Hahn, Germany). The following

mode was used: the volume of the grinding jar: 50 mL; the diameter of the grinding balls: 12 mm in the amount of 6 pieces; the material of the grinding balls: steel; the duration of grinding in the continuous mode: 60 min; the rotation speed: 1000 rpm.

After grinding, the fractional analysis of the resulting TiC powder was performed with help of a photometric sedimentometer FSH-6K (LabNauchPribor, Moscow, Russia). After grinding, the size of the TiC powder was 0.5–0.7 μm with a content of this fraction of at least 80% by weight fraction.

TiC with a dispersion of 0.5–0.7 μm was injected into a metal jet during the mold filling process in an amount of 0.1–0.4% of the metal mass.

### 2.2.2. Melt Preparation and Metal Pouring

The melts were prepared via remelting in an open induction transistor furnace; model UI-25p (MTPC, Mias, Russia); with an improved cooling system. At the end of melting, the melts at a temperature of 1540 °C were poured into corundum crucibles $Al_2O_3$—99% (2FC, Novosibirsk, Russia) with the volume of 120 mL preheated to a temperature of 150 °C.

### 2.2.3. Vibration Treatment of the Melt

Upon completion of casting, the casting molds were subjected to vibration treatment until the end of solidification on an electromechanical vibration stand KD-9363-SPS (Terchy King Design, New Taipei City, Taiwan) with a frequency of 15–60 Hz and an amplitude of 0.5 mm.

### 2.2.4. Melt Treatment Methods

The melt was poured into molds under various modes (Table 1): the calm state, the impact of vibration treatment, the introduction of TiC ultradispersed particles, and the complex effect: vibration and TiC.

### 2.2.5. Determination of the Chemical

The chemical composition of the experimental alloys was determined using a Foundry-Master Lab spectrometer (Hitachi High-Tech, Mannheim, Germany) with an accuracy of 0.01 wt.% for the chemical elements C, Si, Mn, Cr, and Fe, and 0.001 wt.% for S and P. Additionally, the compositions of some samples were determined using energy-dispersive X-ray spectroscopy (EDS, Oxford Instruments, Oxford, UK) on metallographic sections with an accuracy of 0.01 wt.% (Figure 1). For each sample, three areas of $1 \times 1$ mm$^2$ were analyzed.

### 2.2.6. Preparation of Samples for the Study of Mechanical Properties and Structure

To measure the microhardness and tribological properties, and to study the microstructure and cyclic shock dynamic effects, circular cross sections measuring $30 \times 10$ mm (diameter × height) were cut out of the middle part of cylindrical cast blanks with the help of a Unitom-2 manual cutting machine (Struers A/S, Ballerup, Denmark).

Longitudinal sections with a size of $30 \times 5$ mm were cut from the obtained circular cross sections for samples No. 1 and 14 for the study of the microstructure on a scanning electron microscope. The longitudinal sections of samples No. 1 and 14 were embedded in epoxy resin using a Cito Press-1 machine (Struers A/S, Ballerup, Denmark), after which the sections were prepared. The surfaces of the sections were carefully sanded and polished to a mirror finish. Grinding was carried out with a gradual decrease in the grain size of the abrasive paper according to the methodological recommendations of the Metalog Guide (Struers A/S, Ballerup, Denmark). Polishing was carried out on the fabric using a diamond paste. The structure studies were carried out on previously chemically etched sections in a reagent of the following composition: 5 mL of nitric acid (density 1.4 g/cm$^3$), 95 mL of ethyl alcohol.

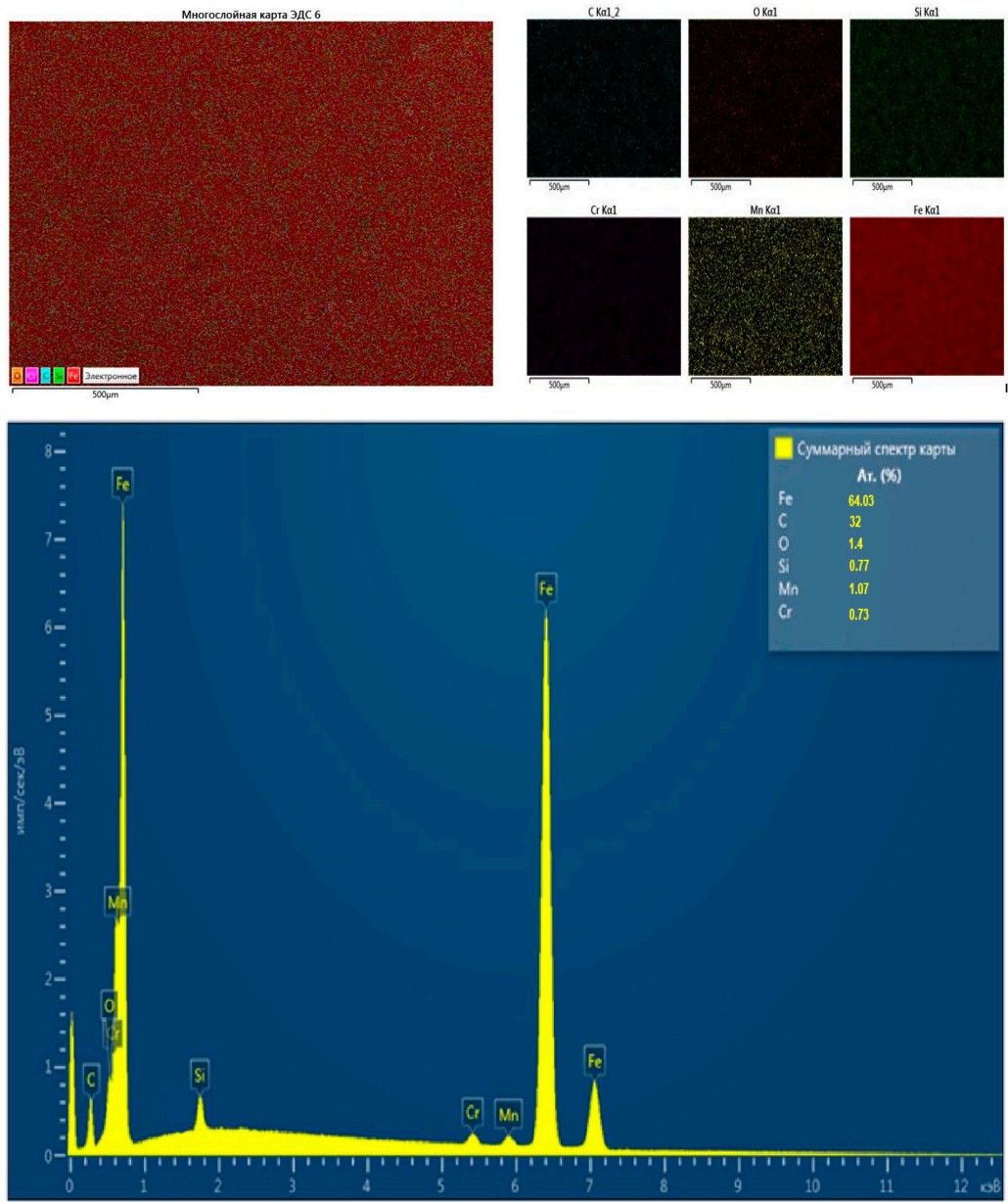

**Figure 1.** SEM (Scanning Electron Microscopy) and EDS (Energy-dispersive X-ray spectroscopy) analyses of the element displaying in the experimental alloy.

Microhardness was tested on unetched sections.

The surfaces of the samples were not polished before the cyclic shock dynamic impact test and the determination of the coefficient of friction.

### 2.2.7. Microhardness Measurement

The Vickers microhardness values were measured on samples using an Emco-Test Durascan 70 hardness tester (Emco-test PrűFmaschinen GmbH, Kuchl, Austria) with the load of 0.5 N and the load application time of 10 s. Eight measurements were taken randomly with the calculation of the arithmetic mean.

### 2.2.8. Microstructure Analysis

The microstructure of the longitudinal sections of the samples was studied using a scanning electron microscope (SEM) JEOL JSM-7600F (JEOL Ltd., Akishima, Tokyo, Japan)

with an OXFORD X-Max 80 detector (Oxford Instruments, Abingdon, UK) in the SEI mode (secondary electrons), power 15 kW at magnification 400, 5000×.

Metallographic studies of the structure of the samples were carried out using a universal metallographic microscope Altami MET 5C (Altami, Moscow, Russia).

Quantitative analysis of the microstructure of the samples was carried out using the Thixomet Pro image analyzer (Thixomet, Saint Petersburg, Russia).

### 2.2.9. Cyclic Impact Measurement

The samples were tested for cyclic shock dynamic impact using the CemeCon Impact Tester (CemeCon AG, Wurselen, Germany). The surface of the samples was subjected to a series of impacts with a ball 5 mm in diameter made of WC–6% Co material under loads of 500 N and 700 N at the constant frequency of 50 Hz. The number of impacts was $10^5$. An optical profilometer WYKO-NT1100 (Veeco Instruments Inc., Plainview, New York, NY, USA) was used to determine the parameters of the wear traces.

### 2.2.10. Tribological Tests

The parameters of wear traces and friction coefficient were studied on an automated friction machine Hightemperature Tribometer (CSM Instruments SA, Peseux, Switzerland) according to the "rod-disk" scheme using one-way rotational motion with a linear speed of 10 cm/s. The test temperature was +25 °C at 70% humidity. A ball 6 mm in diameter made of $Al_2O_3$ was used as a counterbody. The load on the counterbody was 5 N. The coefficient of friction was continuously recorded throughout the entire test cycle of the total distance of 500 m.

## 3. Results and Discussion

### 3.1. Determination of Chemical Composition

The chemical composition of the smelted alloy is shown in Table 2 and Figure 1.

**Table 2.** Chemical composition of the experimental alloy and steel 35HGSL (SS 977-88).

| Alloy | Chemical Element, wt.% | | | | | | |
|---|---|---|---|---|---|---|---|
| | C | Cr | Mn | Si | S | P | Fe |
| Steel 35HGSL (SS 977-88) | 0.3–0.4 | 0.6–0.9 | 1.0–1.3 | 0.6–0.8 | ≤0.04 | ≤0.04 | other |
| Experimental alloy | 0.32 | 0.73 | 1.07 | 0.77 | 0.015 | 0.021 | other |

As can be seen from the data in Table 2, the composition of the experimental alloy corresponds to the steel grade 35HGSL (SS 977-88).

### 3.2. Microhardness Measurement

Figure 2 shows the average values of microhardness dependences for the treatment modes in Table 2.

Figure 2a shows that vibration at a frequency of 15 Hz leads to an increase in hardness to 223 HV (sample 2) compared to the original untreated cast (sample No. 1; 212 HV). With increasing the vibration frequency to 30, 45 Hz (samples No. 5, 8), an increase in microhardness of 227, 231 HV is also observed. The maximum value of microhardness is achieved when the melt is treated with the frequency of 45 Hz, which is 8.23% more than the hardness of the reference (sample No. 1).

A further increase in the processing frequency to 60 Hz led to a decrease in the microhardness values, while liquid metal ejection from the open surface was observed on the melt surface. This release of droplets led to the loss of metal. Clearly, a further increase in frequency is impractical.

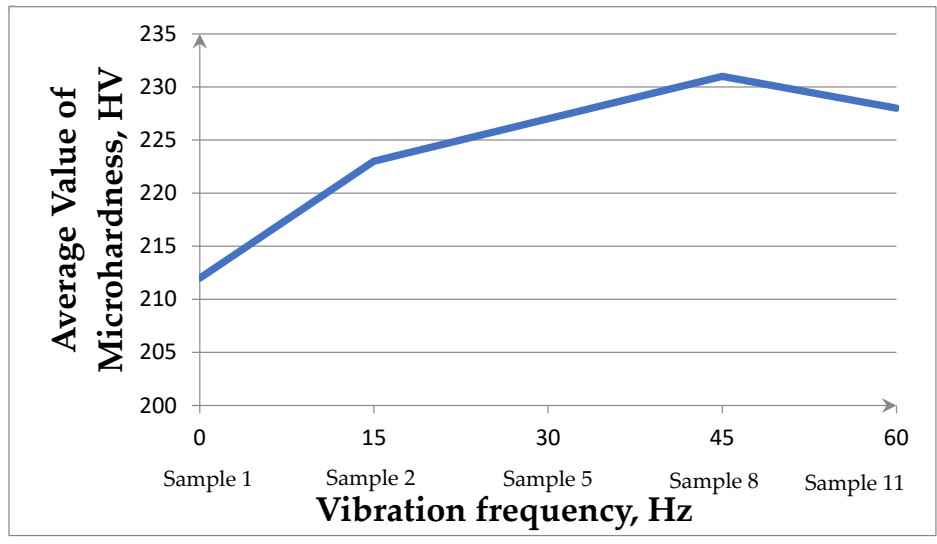

(**a**)

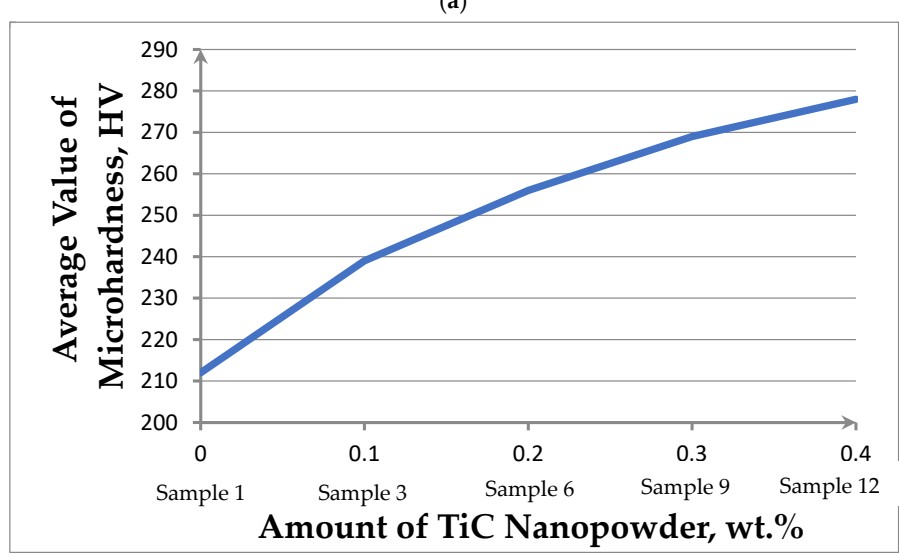

(**b**)

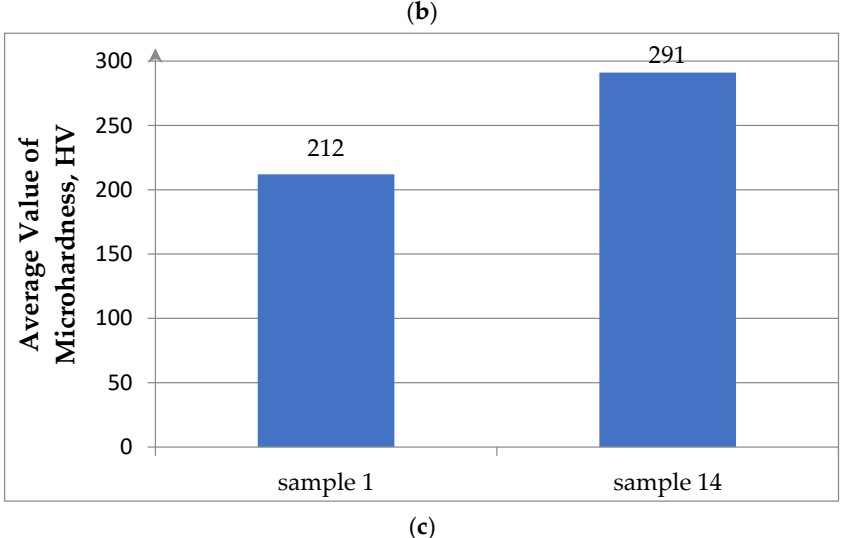

(**c**)

**Figure 2.** Changing average values of microhardness (HV) depending on the treatment mode: (**a**) on the frequency of vibration; (**b**) on the amount of introduced TiC ultradispersed powder; (**c**) on complex treatment.

Adding TiC powder in the amount of 0.1; 0.2; 0.3; 0.4 wt.% increases the alloy microhardness to the following values: 239, 256, 269, 278 HV (Figure 2b). It can be noted that the microhardness values unambiguously correlate with the amount of TiC introduced: the greater the amount of TiC introduced, the higher the hardness.

To enhance the effect of hardening, it was proposed to introduce a complex effect (sample No. 14). In this case, the increase in microhardness was 37.2% compared to sample No. 1 (Figure 2c).

The observed increase in hardness in the course of vibration treatment is clearly associated with changing the conditions of solidification due to the forced mixing of the liquid phase under the conditions of developing the cavitation effect, which provides a significant increase in the number of particles of the solid phase in the melt at the moment of destruction of the solid crust on the surface of the melt [20].

The strengthening effect upon the introduction of TiC can be associated with both structural changes and the formation of a new phase, for example, titanium carbonitrides. To test this assumption, the structures of samples No. 1 and 14 (Table 2) that have the minimum (212 HV) and maximum (291 HV) microhardness values, were studied.

*3.3. Microstructure Analysis*

Figure 3a shows that the cast raw sample No. 1 is characterized by a coarse-grained structure with an average grain diameter of 50 microns, and they are heterogeneous in size. Defects in the microstructure of the casting are also visible in the form of pores and relatively large nonmetallic inclusions (8–13 μm). X-ray microanalysis made it possible to identify these inclusions as manganese sulfides.

In accordance with modern ideas about the sources and dynamics of the development of microstructure defects, it can be assumed that the origin of most types of microdefects is based on mass transfer, which includes the processes of fluid convection and particle movement in the volume of solidifying casting. Consequently, the use of additional forced external physical influence on the liquid melt directly during solidification should be considered as one of the most effective ways to influence the quality of castings [30].

Microstructural studies with image analysis revealed the grinding of grains in the sample after complex processing in the form of vibration and the addition of ultrafine particles of titanium carbide (sample No. 14; Figure 3b). The average diameter of the grains after complex treatment of the melt was 30 μm. This is less than in a cast raw sample.

The alloy subjected to complex processing has the size of nonmetallic inclusions in the range of 1.5–1.8 μm (Figure 3b). These dimensions are smaller than in the cast sample No. 1 (Figure 3a). Nonmetallic inclusions are more evenly distributed in the metal volume compared to the sample without processing. The smaller size of nonmetallic inclusions can probably be explained by the fact that the flows of liquid metal provide erosion of local volumes of melt zones enriched with liquates. For example, sulfides formed under such conditions may be smaller and more evenly distributed over the volume of the crystallizing casting. At the same time, the motion resulting from vibrational stirring reduces the tendency to deposit oxide inclusions already existing in the steel at the phase interface. Thus, vibration creates conditions when, through cavitation, oxide inclusions pass into a protective layer of slag [31].

In addition, it is likely that during vibration, liquid metal flows reduce the size of all growing dendrites, preventing the formation of bridges that cut off liquid volumes of crystallizing liquid. This effect can limit the possibility of shrinkage defects and reduces the porosity of the casting. Thus, it can be assumed that the formation of pores was to some extent prevented by vibration treatment.

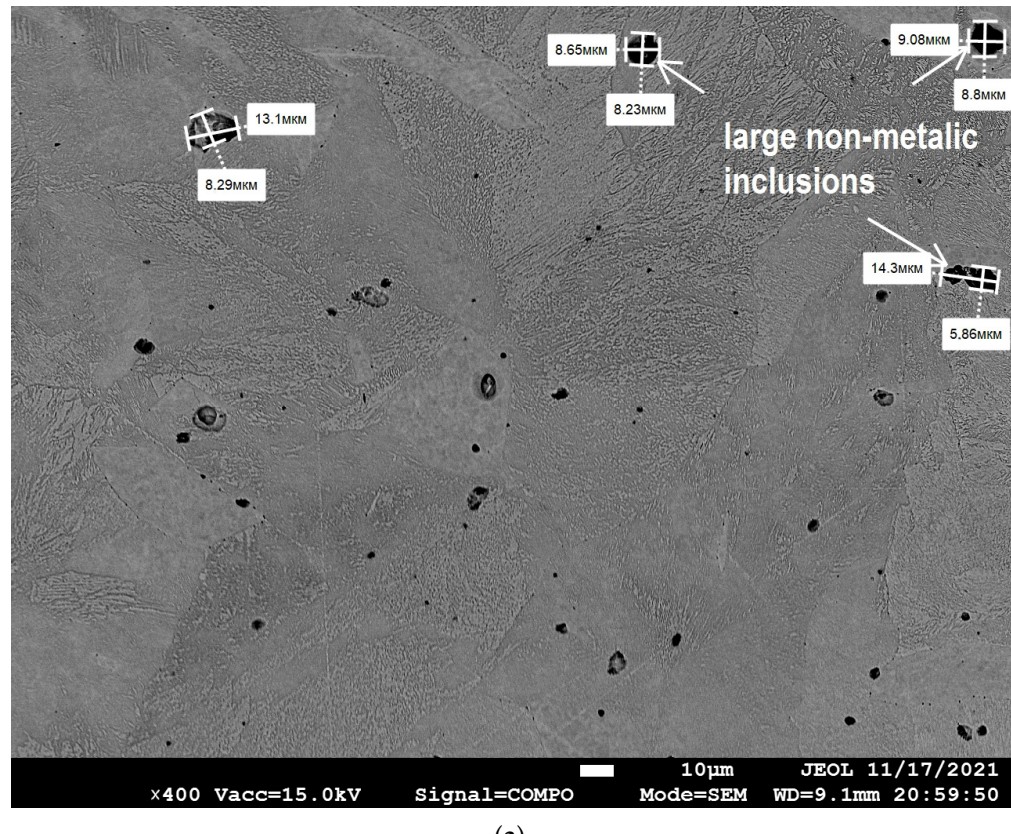

(**a**)

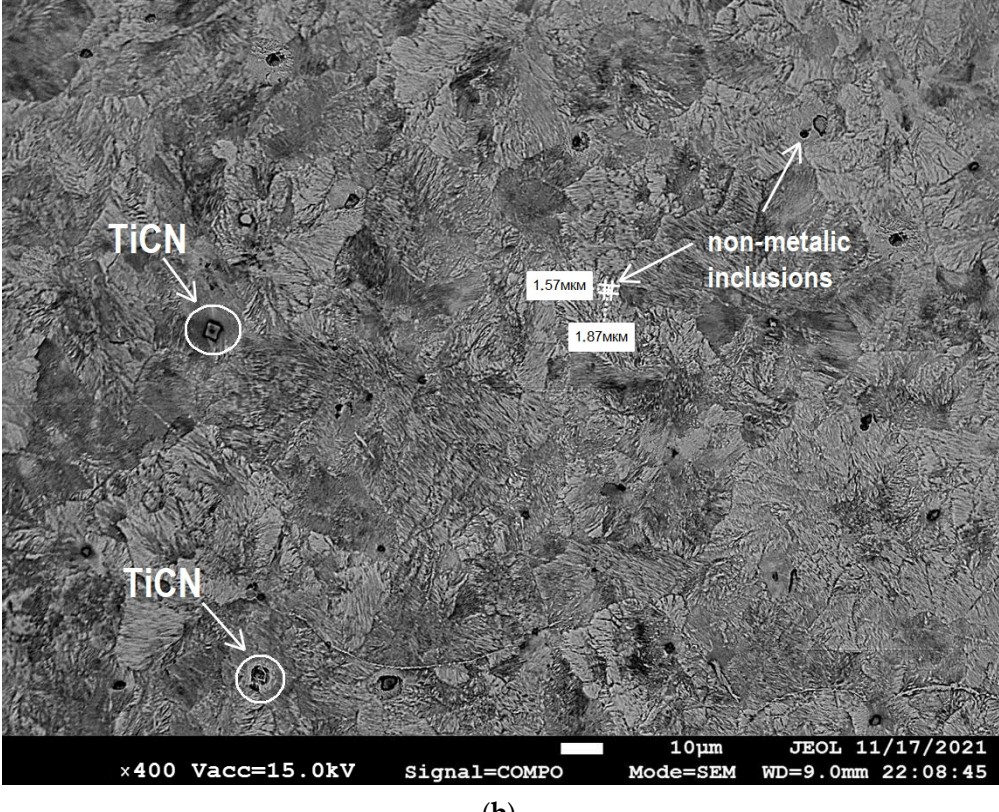

(**b**)

**Figure 3.** Microstructure of steel 35HGSL: (**a**) sample No. 1 without treatment; (**b**) sample No. 14 after complex treatment.

Microstructures were studied at magnifications of ×5000 in order to reveal the effect of grinding metal grains (Figure 4a,b). The investigation showed that the morphology of both the structural components of pearlite colonies in particular and the perlite colony as a whole were changed as a result of complex processing (sample No. 14). Thus, it was found that such an important structural characteristic affecting mechanical properties as the interplate distance in pearlite colonies decreased from a range of 2.6–3.8 μm to a range of 0.48–0.89 μm. At the same time, it can be seen that the thickness of the cementite plates decreased from the range of 1.4–2.0 μm (Figure 4a) to the range of 0.16–0.28 μm (Figure 4b). Additionally, a general grinding of pearlite colonies was observed. This suggested that during the solidification process, the dendrites, which were usually formed in a liquid alloy, were subsequently disrupted and fragmented by the mechanical vibration introduced into the melt. It can be assumed that vibration treatment creates conditions for the destruction of the tops of growing dendrites in the crystallizing melt. Forced melt flows capture crystal fragments and distribute them throughout the volume of liquid metal. Fragments of destroyed dendrites can become additional centers for the generation of new grains [30]. Accordingly, the vibration of the melt can lead to an increase in the number of crystallization centers of austenitic grains and, as a consequence, to a decrease in the grain size of austenite.

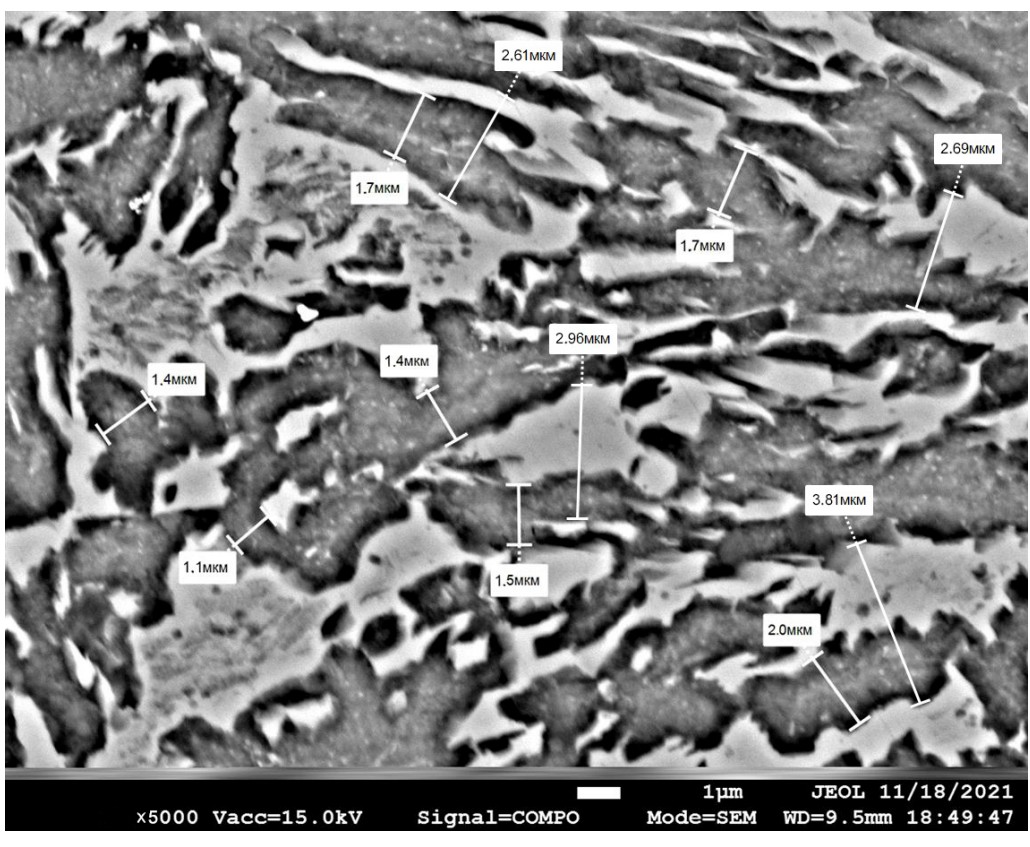

(**a**)

**Figure 4.** *Cont.*

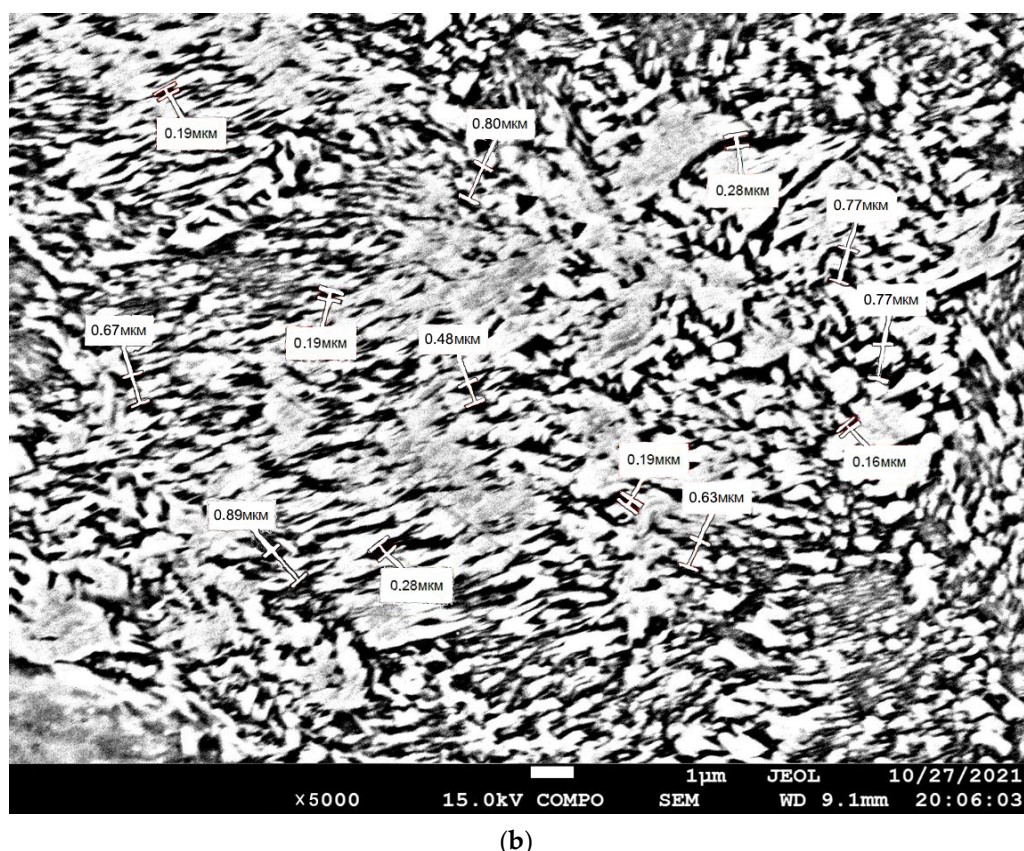

**(b)**

**Figure 4.** The structure of pearlite colonies: (**a**) sample No. 1 without treatment; (**b**) sample No. 14 after complex treatment.

At the same time, part of the dendrite fragments dissolves in the melt. This leads to an intense decrease in the temperature of the liquid metal. Additionally, the introduced refractory titanium carbide particles also lower the melt temperature. A decrease in temperature leads to the decomposition of austenite into the most stable phases—cementite and ferrite.

Cementite nuclei appear in carbon-enriched areas of the γ-phase, mainly at the boundaries of the former austenite grain. The finer the grain of austenite, the greater the total intergranular surface and, accordingly, the greater the total number of nucleation centers of cementite and ferrite. Cementite acquires a lamellar structure due to an increase in the degree of hypothermia. Cementite grows inside the austenite grain. Ferrite crystallizes on the surface of each cementite plate, as on the lining. Alternate crystallization of cementite and ferrite spreads along the boundaries. Colonies of perlite appear, which consist of alternating plates of cementite and ferrite. In addition to the lateral, the end growth of cementite and ferrite plates occurs. The growth of perlite colonies continues until their mutual collision, and is limited by the presence of a large number of barriers in the form of grain boundaries and formed TiCN phases. A large number of small pearlite colonies with a large number of thin plates of cementite are formed as a result.

A more in-depth analysis of the microstructure of sample No. 14 revealed the presence of individual particles of geometric shape in the form of a polyhedron (Figure 5a).

X-ray microanalysis made it possible to identify these inclusions as titanium carbonitride particles (Figure 5b). The presence of these particles confirms the fact that it was possible to effectively introduce solid refractory ultrafine titanium carbide particles into the metal melt and ensure their uniform distribution by volume.

It can be assumed that the formation of the TiCN phase is probably due to the fact that the introduced ultrafine TiC powder participates in a chemical reaction with nitrogen in the metal melt.

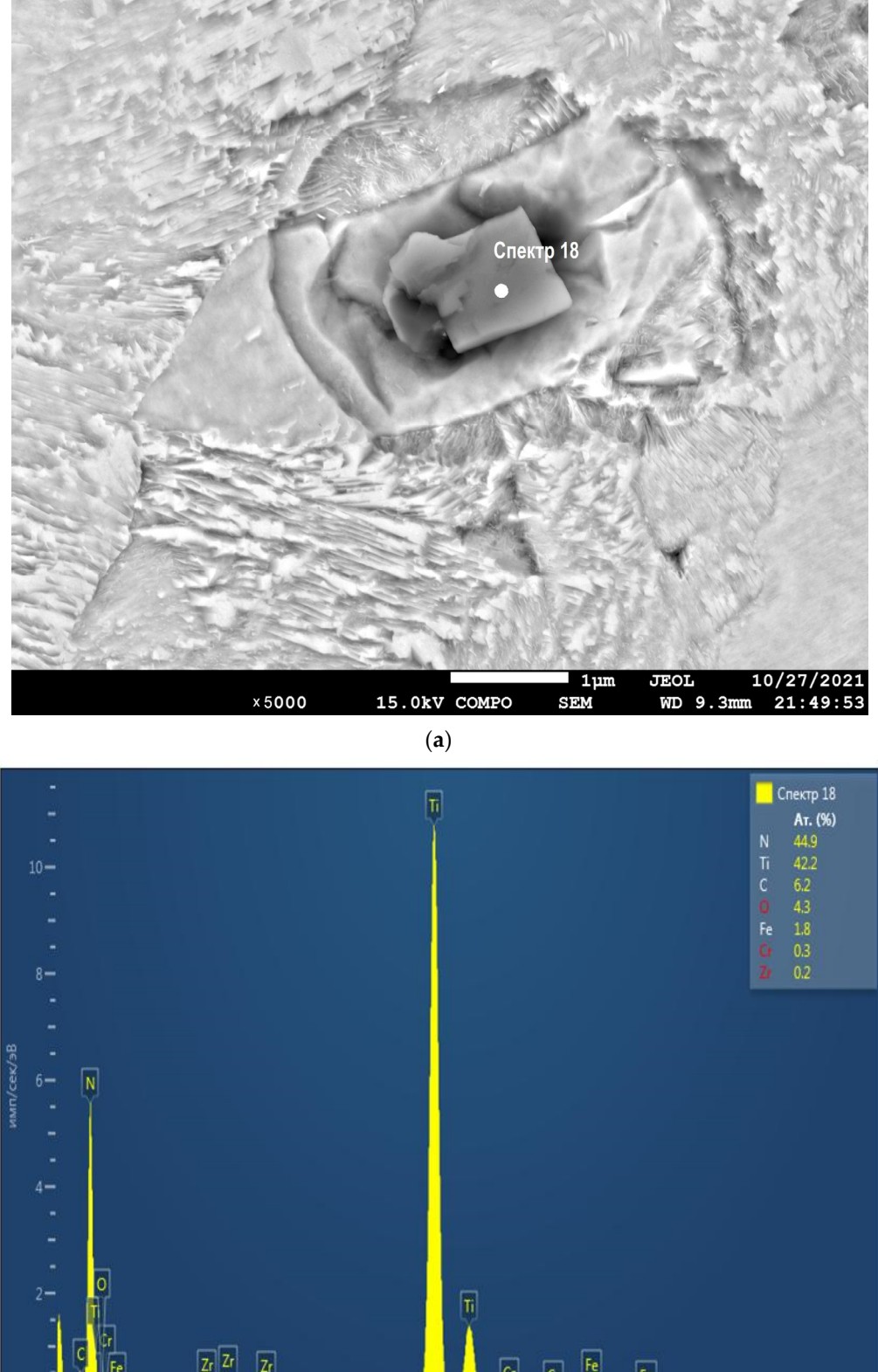

**Figure 5.** Structure containing titanium carbonitride: (**a**) inclusion; (**b**) EDS (Energy-dispersive X-ray spectroscopy) spectrum.

Nitrogen is a harmful impurity found in steel. An increased nitrogen content can lead to the formation of nitrides that appear along the grain boundaries. This can lead to brittle fractures and cracks in the castings. Thus, the binding of nitrogen into a compound in the form of TiCN leads to the neutralization of nitrogen as a harmful impurity [32].

It is clear that the introduction of titanium carbide in the specified amount and vibration treatment lead to positive changes in the microstructure: the grain size decreases, the dispersion of perlite increases, the observed porosity decreases, the size of nonmetallic inclusions decreases, and new phases are formed in the form of titanium carbonitride with a likely decrease in the concentration of free nitrogen in the alloy. Based on this, it can be assumed that the mechanical properties will also change. In order to verify this assumption, the tribological properties were studied.

*3.4. Cyclic Impact Measurement*

It is known that medium-carbon alloy steel in the cast state has a rather low impact resistance due to the fact that the main structural components of the alloy, i.e., ferrite and perlite, have a rough structure and an uneven distribution in volume. In addition, the resistance of medium-carbon alloy steel to impact loads is insignificant due to the large grain shape and the presence of nonmetallic inclusions in the metal matrix, which serve as stress concentrators at impact–dynamic loads. In the process of dynamic action, this leads to the formation of microcracks at the grain boundaries and subsequent fatigue destruction of the metal upon reaching a critical state. The tests for cyclic impact–dynamic action were carried out on selected samples, No. 1, 8, 12, and 14, the processing modes of which are shown in Table 1.

Figure 6 shows the character of the dents on the surface of samples No. 1 and 14 and their profiles after cyclic shock dynamic impact with a counterbody load of 500 and 700 N. Figure 6 demonstrates that sample No. 14 has the greatest resistance, and the lowest parameter belongs to sample No. 1. After testing, the deformation profile on the surface of sample No. 14 has the smallest depth. Visual inspection of the surfaces of samples No. 1 and 14 revealed a difference in the characteristics of the worn surface after testing. The existing differences indicated a difference in the behavior of the metal during cyclic impact wear.

Thus, the surface of sample No. 1 was subjected to severe plastic deformation under the influence of shock stresses. Surface cracks and looseness were found, which were formed under the impact of shock loads on the surface of sample No. 1 (Figure 6a,c). In addition, plastic ridges formed along the perimeter of the print, which did not break away from the base metal. This can indicate a relatively high plasticity of the metal [33].

At the same time, shock loading caused the formation of large chips and the detachment of the surface layer on the surface of the sample No. 14.

The appearance of potholes without the formation of surface cracks, plastic ridges, and looseness can indicate the transformation of the main mechanism from plastic fatigue (sample No. 1) to wear during brittle fracture wear (sample No. 14) [34].

The test results are expressed in the geometric characteristics of the formed craters: the depth Rv; the surface roughness Ra; the diameter b (Table 3, Figure 7).

**Table 3.** Depth Rv, roughness Ra, and diameter b of the craters formed on the surface of the samples.

| Sample No. | 500 N | | | 700 N | | |
|:---:|:---:|:---:|:---:|:---:|:---:|:---:|
| | Rv, μm | Ra, μm | b, mm | Rv, μm | Ra, μm | b, mm |
| 1 | 17.86 | 8.04 | 0.80 | 25.55 | 12.21 | 0.92 |
| 8 | 15.88 | 6.87 | 0.74 | 24.80 | 12.3 | 0.88 |
| 12 | 15.91 | 6.86 | 0.72 | 19.02 | 8.84 | 0.80 |
| 14 | 13.69 | 5.66 | 0.63 | 14.73 | 6.26 | 0.68 |

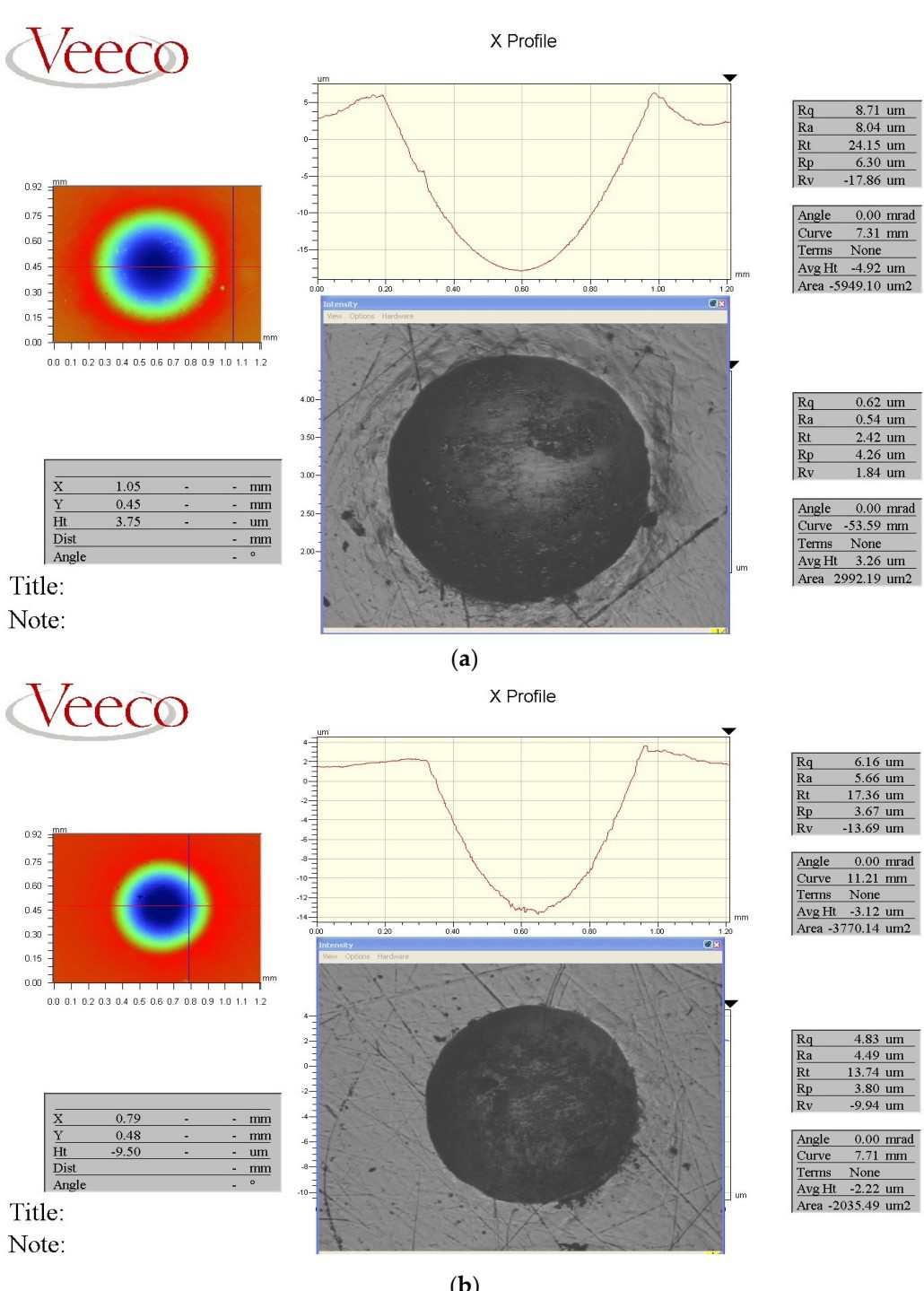

(**a**)

(**b**)

**Figure 6.** *Cont.*

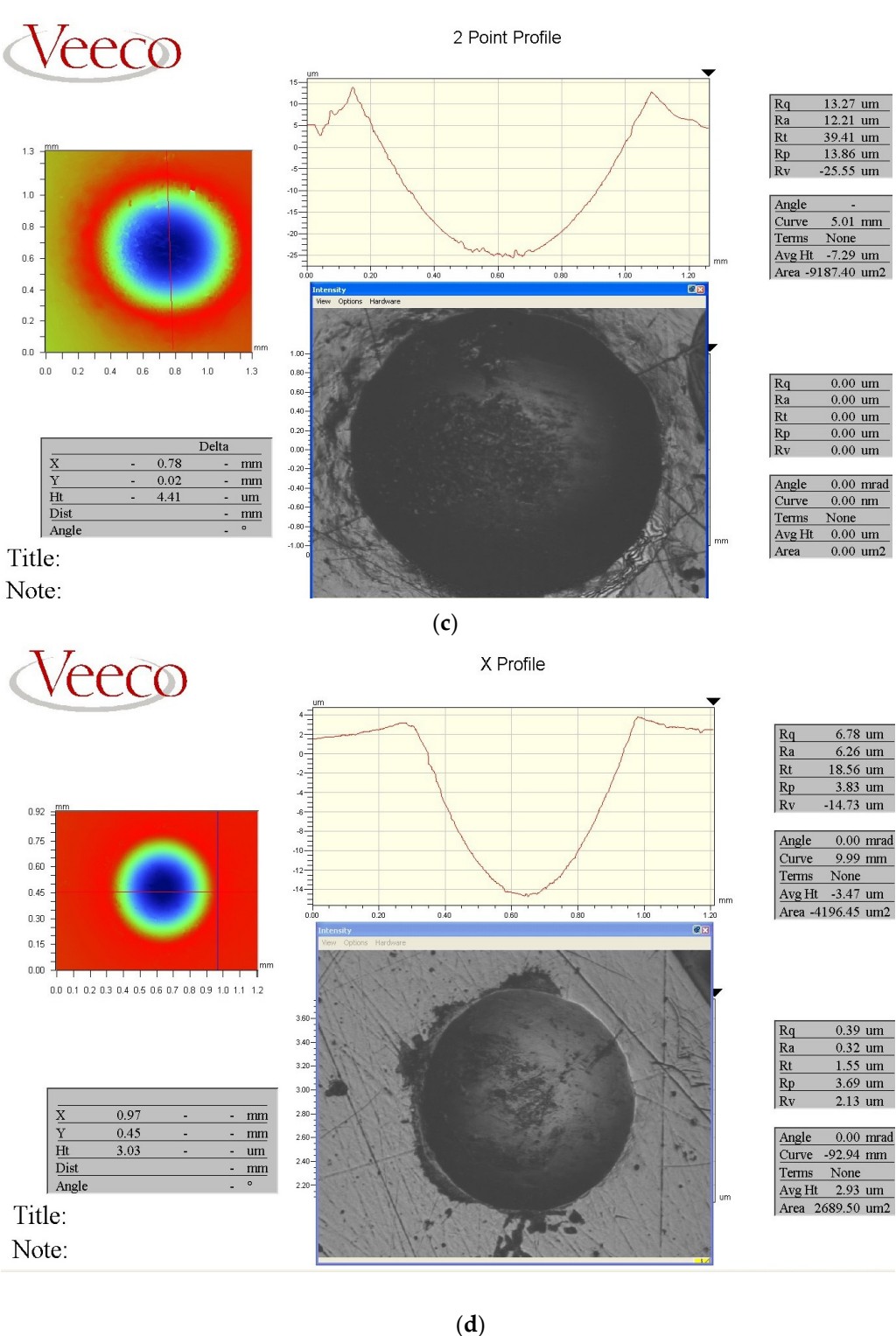

**Figure 6.** Results of testing for shock dynamic impact: (**a**,**c**) sample No. 1 without treatment (500, 700 N); (**b**,**d**) sample No. 14 with complex treatment (500, 700 N).

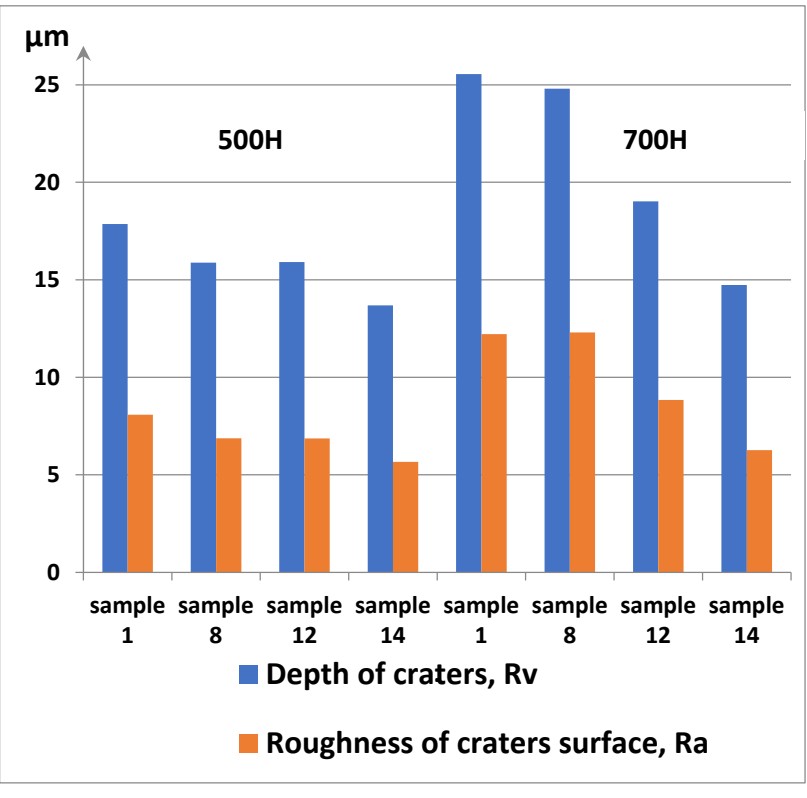

(a)

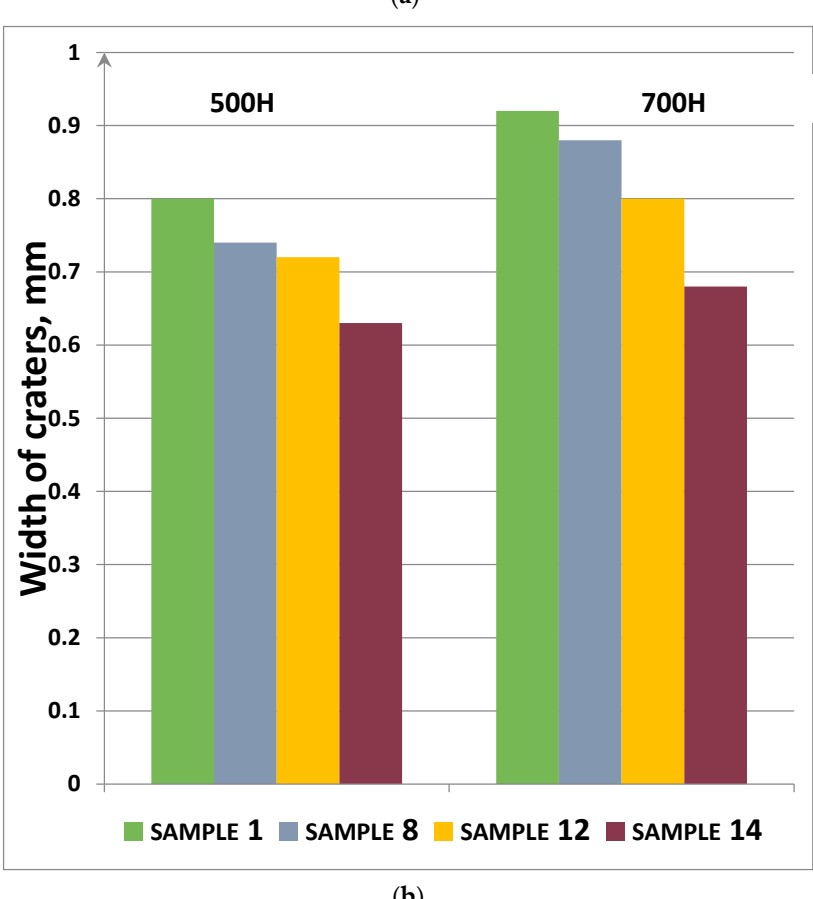

(b)

**Figure 7.** Geometrical parameters of craters on the surface of the samples: (**a**) depth Rv, μm and roughness Ra, μm; (**b**) diameter b, mm.

With the load of 500 N, the smallest counterbody penetration depth was observed for sample No. 14 (13.69 μm), which is 23.3; 10.9 and 11% less than for samples No. 1 (17.86 μm), No.8 (15.88 μm), and No. 12 (15.91 μm), respectively (Figure 7).

The imprint diameter value is minimal for sample No. 14 (0.63 mm), which is 29.27; 22.15 and 14.7% less than for sample No. 1 (0.8 mm), No. 8 (0.74 μm), and No. 12 (0.72 mm), respectively. The maximum depth (25.55 μm) was recorded for sample No. 1, which is 42.34% more than for sample No. 14 (14.73 μm) at 700 N (Figure 7). Figure 8 shows the profile of the craters after testing.

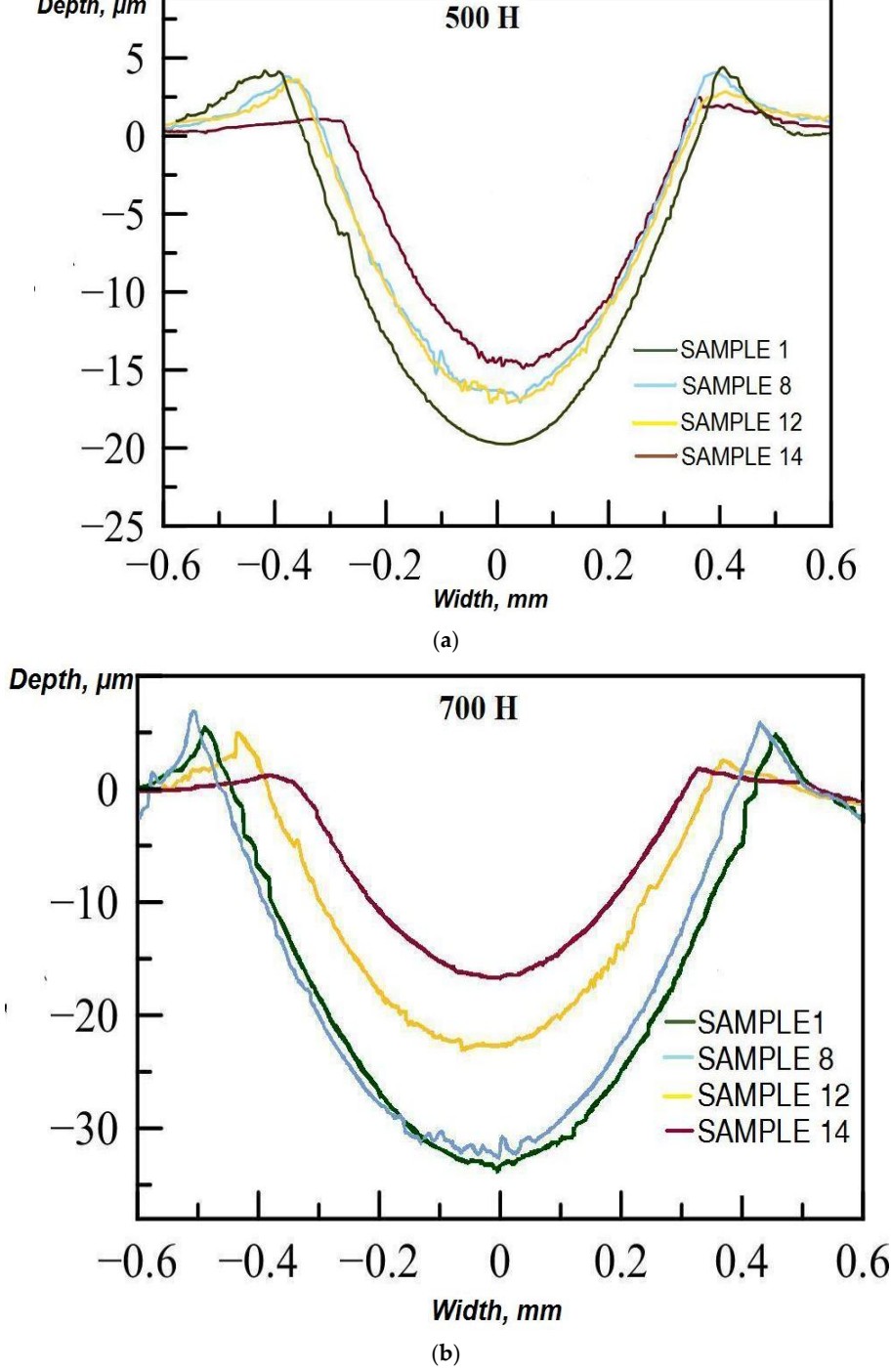

**Figure 8.** Dimensions of crater profiles on the surface of samples after testing. Cyclic shock dynamic impact: (**a**) load 500 N; (**b**) load 700 N.

It can be assumed that the increase in the impact resistance of sample No. 14 is achieved by improving the structure. This improvement is manifested in the form of an increase in the dispersion of perlite colonies (Figure 4b) due to an increase in the degree of supercooling during vibration treatment and the presence of a fine-grained structure by the formation of many additional crystallization centers with the introduction of ultrafine titanium carbide particles (Figure 5a). In addition, large parts of growing dendrites, destroyed by moving fluid flows during vibration, also become additional centers of crystallization.

*3.5. Tribological Tests*

Dry abrasion tests were carried out with help of the high temperature tribometer on samples No. 1, 8, 12, and 14 to study the tribological properties and to understand the frictional interactions of the contact surfaces. The patterns of wear and the profiles of the tested surfaces are shown in Figures 9–11 and Table 4 below. Figure 9 demonstrates the character and parameters of wear traces obtained after abrasion tests as cast and treated in different modes of medium-carbon alloy steel.

The friction forces created by the sliding of the ball on the surface of the test samples led to high deformations occurring in the inner layers of the samples. As the tests were carried out, the zone of plastic deformation under the surface expanded depending on the hardness and the microstructure of the samples. Consequently, the softer cast untreated sample No. 1 had a more significant plastic deformation than the harder samples. This was confirmed by the formation of plastic ridges along the edge of the wear mark of sample No. 1 (Figure 9a). As soon as the microstructures of the samples were subjected to very strong plastic deformation, the nucleation and propagation of cracks under the surface led to the peeling of wear particles from the surface [35]. The samples after processing (No. 8, 12, 14) had better protection against cracking than the softer sample (sample No. 1) due to higher hardness. Consequently, much more debris and obvious signs of wear were formed on the softer sample No. 1 (Figure 10). During the tests, the surface of sample No. 1 was subjected to the microcutting action of the abrasive; as a result, the sample material was removed from the destroyed surface under the action of surface friction forces. On the harder drives, fewer detachments were formed, therefore, less wear residues were formed. The particles that exfoliate from the harder samples act as loose abrasive particles in the tribological system and lead to the minor abrasive wear of the harder samples [36].

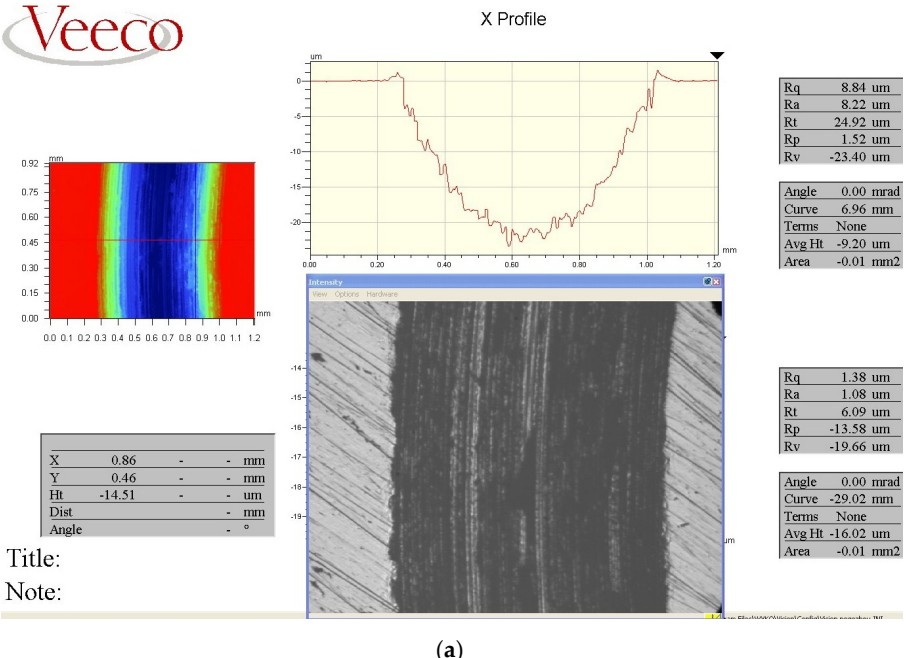

(a)

**Figure 9.** *Cont.*

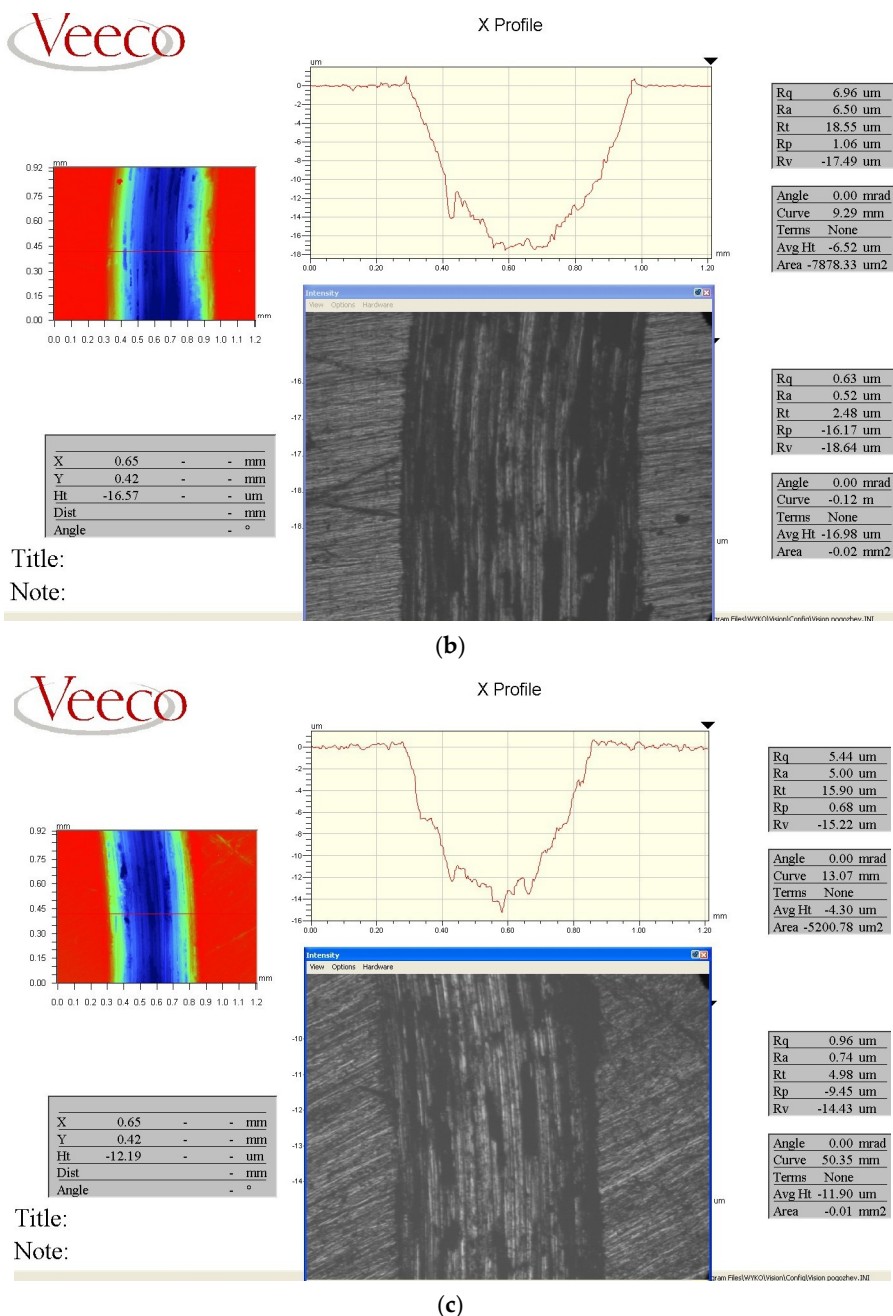

**(b)**

**(c)**

**Figure 9.** *Cont.*

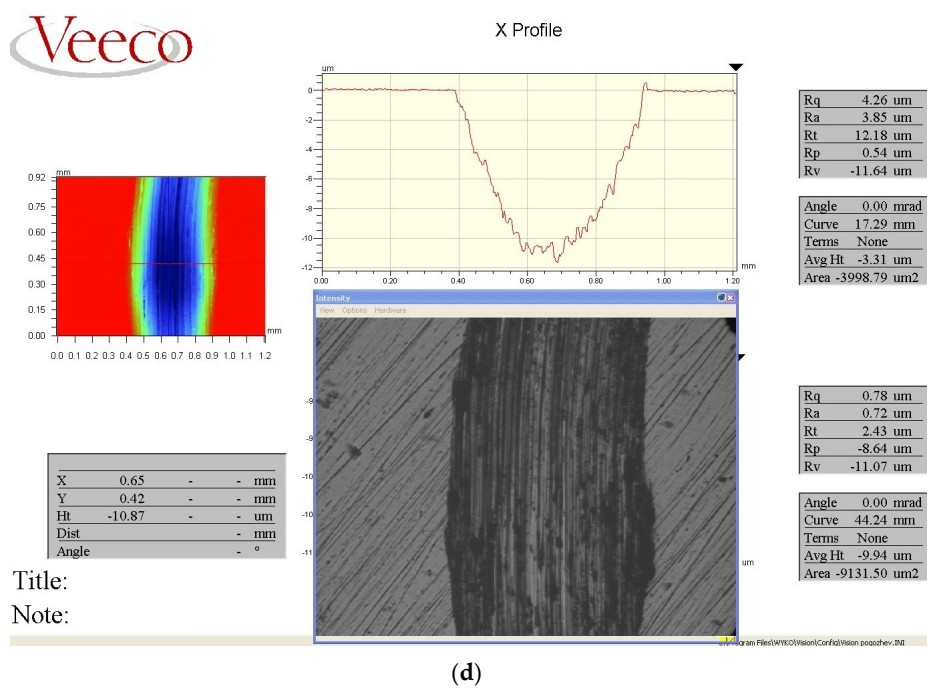

(**d**)

**Figure 9.** Parameters of wear traces after abrasion tests: (**a**) sample No. 1; (**b**) sample No. 8; (**c**) sample No. 12; (**d**) sample No. 14.

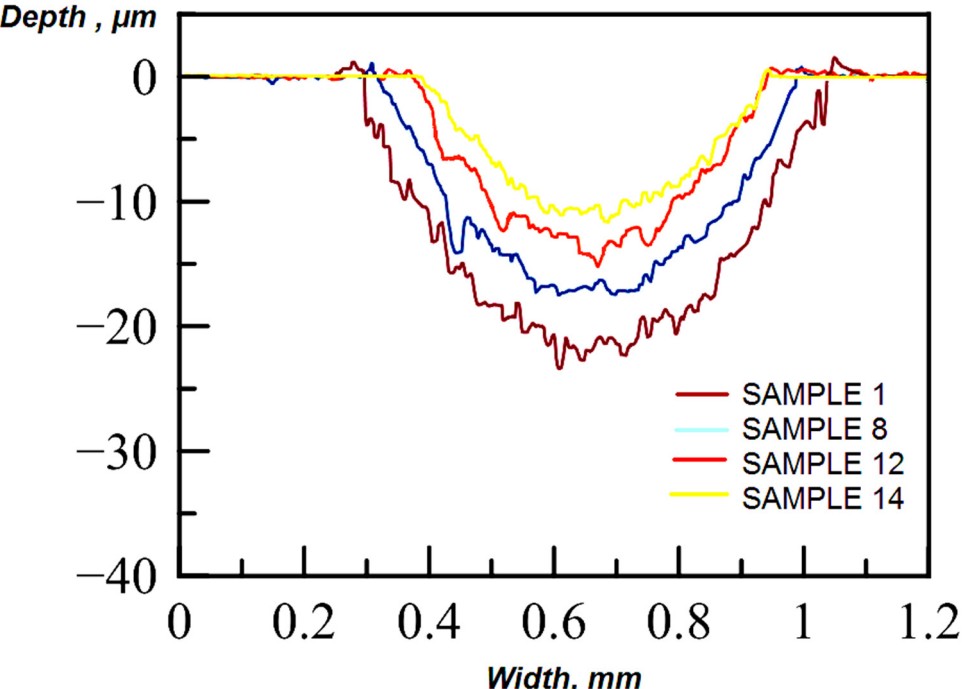

**Figure 10.** Results of abrasion tests: dimensions of wear trace profiles.

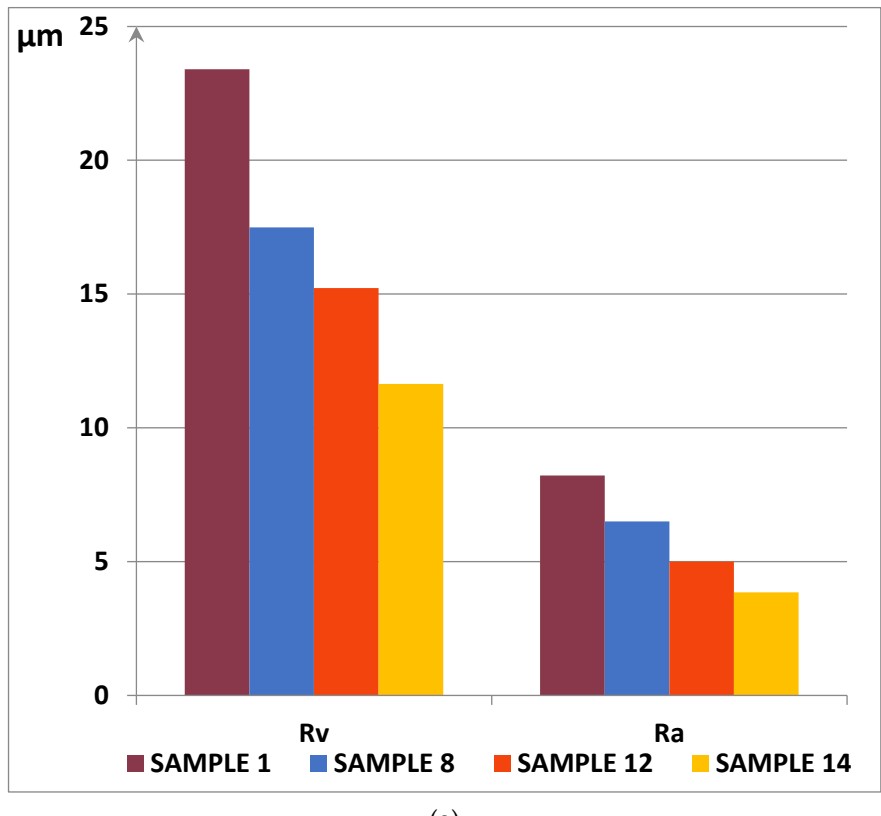

(**a**)

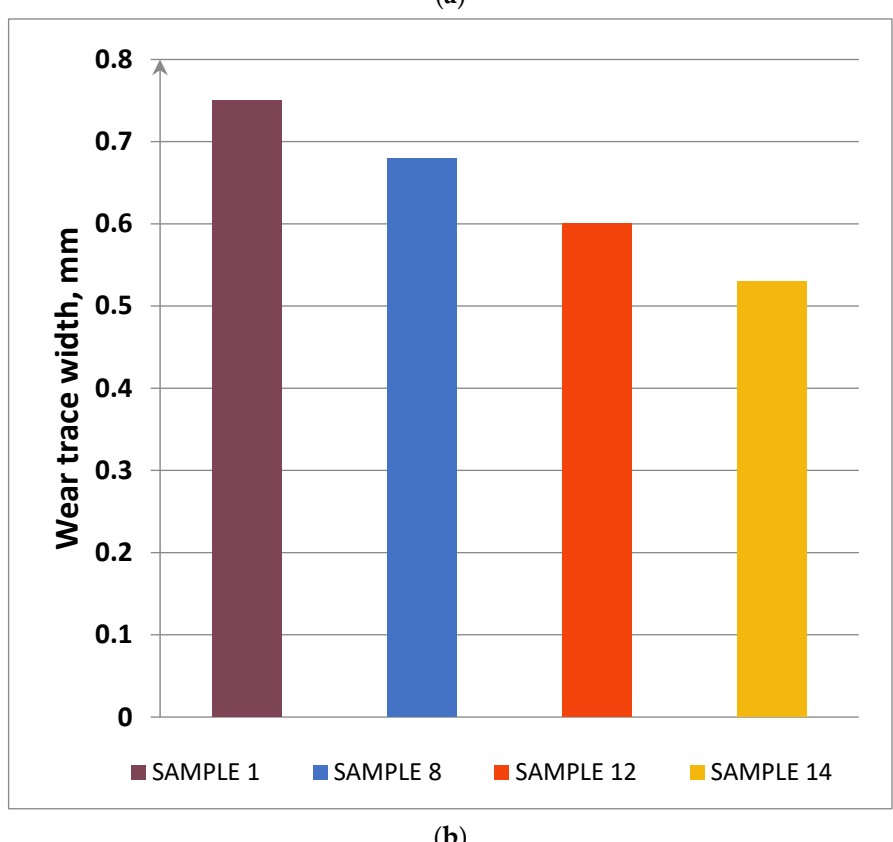

(**b**)

**Figure 11.** Geometric parameters of wear traces: (**a**) depth Rv, μm and surface roughness Ra, μm; (**b**) width b, mm.

**Table 4.** Depth Rv, roughness Ra, and width b of the wear traces.

| Sample No. | Rv, μm | Ra, μm | b, mm |
|---|---|---|---|
| 1 | 23.40 | 8.22 | 0.75 |
| 8 | 17.49 | 6.5 | 0.68 |
| 12 | 15.22 | 5.0 | 0.60 |
| 14 | 11.64 | 3.85 | 0.53 |

Figure 10 shows that the surface wear of samples No. 8, 12, 14 treated with different modes is less than that of sample No. 1, which can be explained by increased hardness values. Sample No. 1 (Figure 10) has the greatest wear, which can be explained by its cast coarse-grained ferrite–pearlite microstructure and the presence of defects in the cast metal base in the form of nonmetallic inclusions. The profile of the traces of sample No. 1 has the form of deep alternating sharp depressions and peaks, which may be a sign of intense destruction. It can be assumed that the surface of sample No. 1 is destroyed in places of discontinuity of the metal base and along the grain boundaries with chips and loss of grains during the tests under the action of dry friction. The minimum value of the width of the wear trace is characteristic of sample No. 14, and the depth of the groove is less than that of samples No. 1, 8, 12. The relief of the worn surface of sample No. 14 is smoother, with the smallest protrusions and depressions, which is noticeable in Figures 9 and 10.

The obtained results of the experiments (Figure 11, Table 4) showed that sample No. 14 has the lowest roughness of all the tested samples. Thus, the surface wear of Ra is 3.85 μm, whereas this parameter is about 8.22 for a cast untreated sample of medium-carbon alloy steel. In turn, other samples have values of this parameter equal to 5.0 (sample No. 8) and 6.5 (sample No. 12). It should be noted that the values of the greatest deepening of the Rv profile as they increase are arranged in the following order: sample No. 14 (11.64 μm); sample No. 12 (15.22 μm); sample No. 8 (17.49 μm); sample No. 1 (23.40 μm). The smallest width of the groove profile at the base b (0.53 mm) also belongs to sample No. 14, to which complex processing was applied, while the highest value for this parameter (0.75 mm) was noted for cast untreated sample No. 1.

The values of the sliding friction coefficients range from 0.7 to 0.83 for the studied samples No. 1, 8, 12, 14. These values are characteristic of the surfaces of materials made similarly to medium-alloy steel obtained by casting without machining. A gradual increase in the sliding friction coefficient in the process of increasing the test distance (Figure 12) indicates a sufficiently intense nature of surface wear. When the particles destroyed and separated from the surface of the sample, having high hardness, large fraction, and sharp shape had a noticeable abrasive effect on the surface. It can be assumed that, with an increase in the wear distance, the dominant wear mechanism changed from minor plastic deformation to microcutting and adhesive wear.

The results obtained allow us to identify an obvious pattern between the values of the coefficient of friction, the roughness of the worn surface and the degree of its wear, depending on the type of sample processing. We consider it appropriate to use the data of visual inspection of deformation foci and geometric parameters of the relief of grooves on worn surfaces to assess the effect of the type of processing on the wear-resistant properties of medium-carbon alloy steel. Complex processing of the alloy (sample No. 14) resulted in lower wear losses compared to the cast untreated alloy (sample No. 1). In our opinion, this is due to the higher hardness of the material of sample No. 14, the structure of which was finer-grained compared to the cast raw sample.

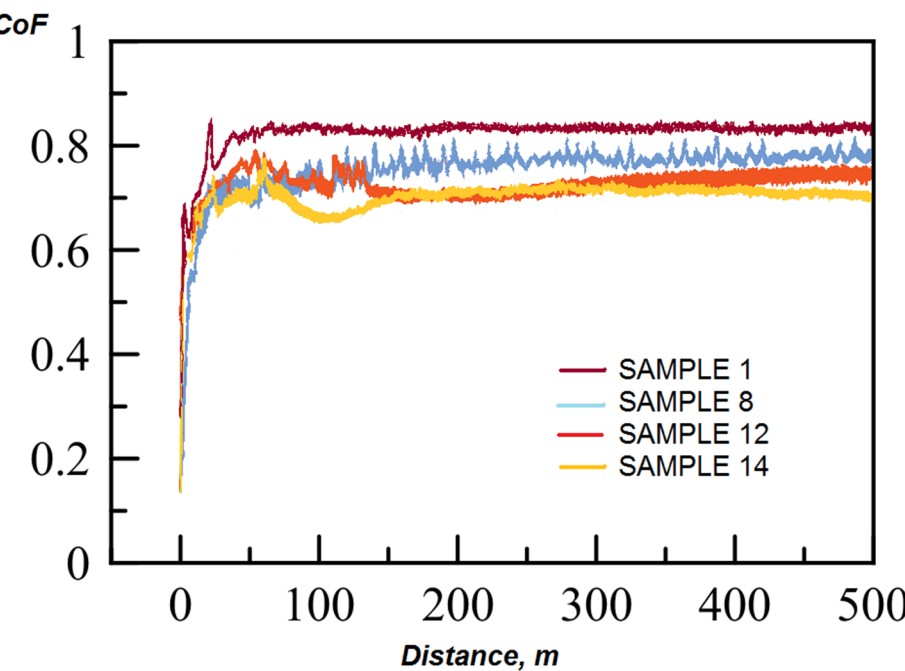

**Figure 12.** Results of abrasion tests: the friction coefficient dependence on the distance.

### 4. Conclusions

Steel casting without additional treatment can be characterized by a coarse-grained structure containing casting defects in the form of pores and nonmetallic inclusions. It is known that the dispersion of the structure and the formation of defects largely depend on the conditions of solidification.

To improve the quality of the metal, an extra furnace treatment was proposed in the form of the introduction of ultrafine titanium carbide particles with simultaneous vibration treatment. The essence of the treatment was a purposeful effect on the liquid melt at the stage of its crystallization.

Apparently, the effect of vibration treatment is manifested in the endogenous mechanism of the formation of new crystallization centers, probably due to the melt flows destroying the growing dendrites during cavitation. Some fragments of dendrites become new embryos for grain growth, and there is a thinning of the structure. Thus, the average grain diameter decreased to 30 μm compared to the untreated cast sample (50 μm). Additionally, a part of the destroyed dendrites melts, which leads to the supercooling of the melt. With an increase in the supercooling of austenite, the rate of nucleation of the centers of pearlite colonies increases faster than the linear rate of their growth. This leads to a decrease in the size of the colonies. This is important for increasing the strength of the metal, since pearlite colonies behave like independent grains when steel is destroyed. Such parameters of pearlite colonies as the interplate distance decreased from a range of 2.6–3.8 μm to a range of 0.48–0.89 μm and the thickness of cementite plates decreased from a range of 1.4–2.0 μm to a range of 0.16–0.28 μm.

Taking into account the size of the introduced titanium carbide particles (0.5–0.6 μm), it can be assumed that these particles are not effective independent crystallization centers. The hardening mechanism of the introduced ultrafine titanium carbide powder appears to be manifested in the following ways. Firstly, the refractory powder particles introduced into the melt cause the supercooling of the melt during crystallization. This leads to the grinding of the grain size of austenite. Secondly, the introduced refractory particles, deposited along the boundaries of growing grains, are barriers to crystal growth, suppressing the formation of a coarse-grained structure. Thirdly, the highly hard refractory TiCN phases distributed over the volume of the metal matrix, which were formed after the introduction of ultrafine TiC powder particles, prevent the introduction of abrasive particles.

The result of the change in the structure was an increase in microhardness values by 37.2% for samples after processing.

Due to the different microhardness, the main wear mechanism of the control samples differed under the same conditions of the impact wear test. The wear mechanism changed from plastic fatigue wear to wear during brittle fracture.

During the dry friction test, with an increase in the wear distance, the dominant wear mechanism of the tested samples varied from slight plastic deformation to microcutting and adhesive wear.

It was found that the effect of vibration and ultrafine particles of titanium carbide led to the grinding of macro- and microstructures; as a consequence, this led to a decrease in wear parameters for samples after processing.

**Author Contributions:** Conceptualization, T.K. (Tatyana Kovaleva); methodology, T.K. (Tatyana Kovaleva); validation, S.K. (Svetlana Kvon); resources, Y.S. (Yevgeniy Skvortsov) and A.S. (Anna Skvortsova); writing—original draft preparation, Y.S. (Yevgeniy Skvortsov); writing—review and editing, Y.S. (Yevgeniy Skvortsov) and S.K. (Svetlana Kvon); visualization, A.S. (Anna Skvortsova); supervision, S.K. (Svetlana Kvon), M.G. (Michot Gerard) and A.I. (Aristotel Issagulov); project administration, M.G. (Michot Gerard) and V.K. (Vitaly Kulikov); funding acquisition, V.K. (Vitaly Kulikov). All authors have read and agreed to the published version of the manuscript.

**Funding:** This research was funded by the Science Committee of the Ministry of Science and Higher Education of the Republic of Kazakhstan (Grant No. AP15473207) "Development of technology for the manufacture of defect-free homogeneous castings by casting on gasified models".

**Institutional Review Board Statement:** Not applicable.

**Informed Consent Statement:** Not applicable.

**Data Availability Statement:** All the data presented and/or analyzed in this study are available upon request from the corresponding author.

**Conflicts of Interest:** The authors declare no conflict of interest.

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
