# Peer review of "Titanium Carbide and Vibration Effect on the Structure and Mechanical Properties of Medium-Carbon Alloy Steel"

_coatings, doi:10.3390/coatings13071135_

Round 1

Reviewer 1 Report

Dear Authors,

I appreciate your research, but in my opinion an improvement of the presentation quality of your results is mandatory.

First a clear distinction between materials and methods and results and discussion needs to be established. In the materials part the chemical composition of the alloy is presented, with an EDS spectra and elemental mapping. Although your decision to place it here is understandable, I would like to see these results in the proper section, i.e. Results and Discussion. 

The treatment modes presented in table 2 are somewhat confusing and need explaining. Sample 1, 4, 7, 10 are the same - they are used as reference samples thus four distinct batches were prepared? If so, why only sample 1 is used as reference? The rest of sample coding is clear, check for sample 14, where the frequency is most likely 60Hz, not 45Hz as presented.

The quality of the charts is unsatisfactory, and lack information. The standard deviation needs to be placed on charts / reported somewhere. As an example, in fig. 2, place error bars for average hardness values, and a label would be helpfull to identify batch number. 

Please revise the statement in line 202 regarding the rhombus shape of the particle - using SEM its polyhedral shape is obvious.

In fig. 5 the screen captures presented as results are not acceptable, in my opinion. The numerical values in the tables are unreadable due to low resolution. Present only the images with higher resolution, if possible.

Same observations for figure 8.

I think the paper would benefit of a grain size analysis, which would sustain your results and inferrences. 

Please revise your manuscript, there are several formulations that need revison, as an example the phrase in line 54-55, line 137 "load exposure time", line 154 - I think it is Tribometer, not Tribometr, line 192 - visual observation (I do not understand).

My best regards.

Reviewer 2 Report

The manuscript "Titanium Carbide and Vibration Effect on the Structure and Mechanical Properties of Medium-carbon Alloy Steel" reports an effective method for addition of TiC to a steel matrix. The study aimed to improve the hardness and wear behavior of a medium-carbon alloy steel through the addition of titanium carbide nanopowder and the low-frequency vibration treatment during solidification. Authors postulated that the complex effect of low-frequency vibration with the additional introduction of a small amount of titanium carbide nanopowder with the size of 0.5-0.7 μm during the casting process has a positive effect on structural changes and leads to increasing mechanical properties. Authors reported a 37.2 % increase in the mechanical properties. In terms of data, this paper is indeed very rich. I recommend this manuscript for publication after following mandatory revisions.

1.     This paper needs a major revision on the strengthening mechanisms. To me, it is more a report of several data and analysis. Yet, the discussion on the strengthening mechanisms, whether it is the direct consequence of particle addition, or change in the grain structure, is rather poor. The following articles on the “strengthening mechanisms” are highly recommended to be used, as they contain relevant discussions on the topic (though not directly related to your alloying system, the concepts are applicable):

-        Zhang, Z., Yang, Q., Yu, Z., Wang, H., & Zhang, T. (2022). Influence of Y2O3 addition on the microstructure of TiC reinforced Ti-based composite coating prepared by laser cladding. Materials characterization, 189. doi: 10.1016/j.matchar.2022.111962

-        Zhang, B., Wang, Z., Yu, H., & Ning, Y. (2022). Microstructural origin and control mechanism of the mixed grain structure in Ni-based superalloys. Journal of Alloys and Compounds, 900, 163515. doi: https://doi.org/10.1016/j.jallcom.2021.163515

-        Deng, H., Chen, Y., Jia, Y., Pang, Y., Zhang, T., Wang, S., Yin, L. (2021). Microstructure and mechanical properties of dissimilar NiTi/Ti6Al4V joints via back-heating assisted friction stir welding. Journal of Manufacturing Processes, 64, 379-391. doi: https://doi.org/10.1016/j.jmapro.2021.01.024

2.     Are “titanium carbide nanopowder with the size of 0.5-0.7 μm” really nano-sized particles? Shouldn’t nano-sized particles be smaller than 100 nm?

3.     The quality of most images very poor. For instance, I see nothing in Fig. 1 (elemental mapping).  

4.     If possible, provide a SEM image from as-received particles.

5.     Figure 2 should be reported with error bars. Please explain exactly how you have calculated errors in your measurements.

6.     Can you make a quantitative statement on the grain-size? Have you measured average grain-size in different samples? Generic statements like this one “The structure of the sample after complex treatment (sample â„– 14) is characterized by finer grains” are probably not appropriate when it comes to describing a microstructure.

7.     I am not sure in such large particles (the one you have shown in Figure 4a in 1 um) can have much positive contribution to the mechanical properties, as authors have stated: “new strengthening phases are formed in the form of titanium carbonitride”. This brings me back to my first comment on the necessity of adding a discussion on the strengthening mechanisms.

8.     In line 251-253, authors have stated “high gas porosity, the presence of large non-metallic inclusions (Figure 2), which, as a result, reduces the density and, as a result, the wear resistance of the material”. This gas-porosity needs a further clarification. Does it have to do with the vibration? Moreover, these large particles are postulated to deteriorate wear resistance of the alloy. Doesn’t this contradict previous positive statements over the mechanical properties?

9.     The conclusion is very short and generic.

10.  Absolutely nothing is said about the wear mechanisms. Please add a few lines on the wear mechanisms.  

Reviewer 3 Report

1.     One of the keywords “modifying” should be removed.

2.     The quality of Figures in the whole manuscript should be improved.

3.     The literature review should be elaborated. For example, there are few cited publications in the past three years.

4.     In Section 2, the experimental procedures should be introduced in detail.

5.     In Section 3, the detailed discussion for the results of Figure 10 should be provided.

6.     Please add some contents on the potential applications for the studied materials.

7.     The main contribution of this work should be concise.

8.     From the viewpoint of metallurgy, please explain the titanium carbide and vibration effect on mechanical properties of medium-carbon alloy steel.

 I think one sentence cannot be one paragraph, please reorganize the corresponding text. Some typo mistakes should be corrected. 

Round 2

Reviewer 2 Report

Dear authors, thank you for your effort in the revision. I assume a wrong "reply to comments" file has been uploaded for, since non of these comments are mine. But, looking at the revised manuscript, my main three comments are still not fully addressed. I put them here again for your consideration. Please make sure you have more "discussions". Otherwise, this paper to me is more like a technical report. Also, I am not sure if you have updated the literature list. I see some changes in the text, but nothing in references.

Comments No. 1: This paper needs a major revision on the strengthening mechanisms. To me, it is more a report of several data and analysis. Yet, the discussion on the strengthening mechanisms, whether it is the direct consequence of particle addition, or change in the grain structure, is rather poor. The following articles on the “strengthening mechanisms” are highly recommended to be used, as they contain relevant discussions on the topic (though not directly related to your alloying system, the concepts are applicable):

-        Zhang, Z., Yang, Q., Yu, Z., Wang, H., & Zhang, T. (2022). Influence of Y2O3 addition on the microstructure of TiC reinforced Ti-based composite coating prepared by laser cladding. Materials characterization, 189. doi: 10.1016/j.matchar.2022.111962

-        Zhang, B., Wang, Z., Yu, H., & Ning, Y. (2022). Microstructural origin and control mechanism of the mixed grain structure in Ni-based superalloys. Journal of Alloys and Compounds, 900, 163515. doi: https://doi.org/10.1016/j.jallcom.2021.163515

-        Deng, H., Chen, Y., Jia, Y., Pang, Y., Zhang, T., Wang, S., Yin, L. (2021). Microstructure and mechanical properties of dissimilar NiTi/Ti6Al4V joints via back-heating assisted friction stir welding. Journal of Manufacturing Processes, 64, 379-391. doi: https://doi.org/10.1016/j.jmapro.2021.01.024

Comments No. 2:  I am not sure if such large particles (the one you have shown in Figure 4a in 1 um) can have much positive contribution to the mechanical properties, as authors have stated: “new strengthening phases are formed in the form of titanium carbonitride”. This brings me back to my first comment on the necessity of adding a discussion on the strengthening mechanisms.

Comments No. 3:  I am not sure if such large particles (the one you have shown in Figure 4a in 1 um) can have much positive contribution to the mechanical properties, as authors have stated: “new strengthening phases are formed in the form of titanium carbonitride”. This brings me back to my first comment on the necessity of adding a discussion on the strengthening mechanisms.

Reviewer 3 Report

The revised manuscript can be accepted now.

Round 3

Reviewer 2 Report

I would like to thank authors for the revision. 

The paper is acceptable.